# Risk of Fall in Patients with Functional Hallux Limitus: A Case–Control Study Using an Inertial Measurement Unit

**DOI:** 10.3390/bioengineering12101094

**Published:** 2025-10-10

**Authors:** Jorge Posada-Ordax, Marta Elena Losa-Iglesias, Ricardo Becerro-de-Bengoa-Vallejo, Eduardo Pérez-Boal, Bibiana Trevissón-Redondo, Israel Casado-Hernández, Vicenta Martínez-Córcoles, Anna Sánchez-Serena, Eva María Martínez-Jiménez

**Affiliations:** 1Departamento de Enfermería y Fisioterapia, Universidad de León, Av de Astorga s/n, 24401 Ponferrada, León, Spain; jposo@unileon.es (J.P.-O.); btrer@unileon.es (B.T.-R.); 2Departamento de Enfermería y Estomatología, Universidad Rey Juan Carlos, Av Atenas s/n, 28922 Alcorcón, Madrid, Spain; marta.losa@urjc.es; 3Departamento de Enfermería, Facultad de Enfermería, Fisioterapia y Podología, Universidad Complutense de Madrid, Av Ramón y Cajal s/n, 28040 Madrid, Spain; ribebeva@ucm.es (R.B.-d.-B.-V.); isracasa@ucm.es (I.C.-H.); evamam03@ucm.es (E.M.M.-J.); 4Grupo de Investigación FEBIO, Av Complutense sn, 28040 Madrid, Spain; 5Department of Behavioral Sciences and Health, Miguel Hernández University of Elche, 03550 San Juan de Alicante, Spain; 6Department of Podiatry, Faculty of Health Sciences at Manresa, Universitat de Vic-Universitat Central de Catalunya (UVic-UCC), Av. Universitària, 4-6, 08242 Manresa, Spain; asanchez03@umanresa.cat

**Keywords:** functional hallux limitus, fall risk, mobility, postural stability, inertial measurement unit

## Abstract

Functional hallux limitus (FHL) is a biomechanical condition defined by restricted motion of the first metatarsophalangeal joint during walking, which may impair stability and increase fall risk in older adults. This study compared fall risk between patients with asymptomatic FHL and healthy controls using validated assessments. The case–control design included 40 participants over 65 years, divided into 20 with FHL and 20 controls. Mobility was evaluated with the Timed Up and Go Test, postural stability with the Berg Balance Scale, and fear of falling with the Falls Efficacy Scale—International (FES-I). Spatiotemporal gait parameters were measured using an inertial measurement unit (IMU). No significant differences were found between groups in the Timed Up and Go Test (*p* = 0.694), Berg Balance Scale (*p* = 0.903), Falls Efficacy Scale—International (*p* = 0.913), or spatiotemporal parameters. These results suggest that asymptomatic FHL does not significantly affect mobility, stability, or fear of falling in older adults, indicating that it is not a determining factor for fall risk under controlled conditions. Further research is needed in less controlled settings or in patients with painful FHL.

## 1. Introduction

The risk of falls is a significant public health challenge, as they are one of the leading causes of severe injuries in older adults [1]. Falls account for 40% of injury-related deaths and cause between 20% and 30% of injuries ranging from mild to severe, including soft tissue damage and fractures, particularly in the elderly population [2,3]. This issue is further exacerbated by a high fall prevalence of approximately 25% among older adults, who may experience at least one fall per year [4]. Therefore, it is essential to identify fall-related and balance-related risk factors early on, with the aim of developing effective clinical prevention strategies [5].

Over the years, the use of advanced technologies in the development of fall detection systems has been promoted, such as inertial measurement units (IMUs) [6,7,8,9,10,11,12], instrumented insoles [13], smartphones applications [14,15,16], and optoelectronic systems [17]. These systems, combined with validated questionnaires and tests [18], have improved the predictive capacity of fall detection tools, significantly enhancing the identification of fall risk [19].

Furthermore, it has been observed that some foot pathologies may act as fall risk factors [20], and several foot and ankle components—such as alignment, range of motion (ROM), and muscle strength—are closely linked to balance [21,22]. In this context FHL has also been investigated through the analysis of dynamic plantar pressures using baropodometric platforms, showing an alteration in the distribution of plantar pressures during gait. These changes result in inefficient propulsion, reflecting the biomechanical impact of FHL on foot function and its potential influence on stability and locomotion [23]. FHL is a pathology characterized by restricted motion of the first metatarsophalangeal joint during gait, which can significantly alter foot biomechanics or dynamic activities, even though the range of motion may appear normal during static and non-weight-bearing evaluation [24]. This condition also affects foot biomechanics by reducing the ability to generate efficient propulsion during walking [25].

For this reason, the aim of this study was to compare the risk of falls in patients with FHL and healthy subjects using the validated Falls Efficacy Scale-International (FES-I) [26,27,28,29], the validated Berg Balance Scale (BBS) [30,31], and the validated Timed Up and Go Test (TUG) [32], implemented with an inertial measurement unit previously used to assess TUG with high accuracy and repeatability [33,34,35]. Considering this, the cited study has confirmed that the TUG is a reliable and reproducible tool, recommended in clinical guidelines for its simplicity, quick administration, and usefulness in assessing functional mobility in older adults. However, these same studies warn that its isolated predictive capacity for fall risk is limited, and therefore it should not be used as the sole criterion to identify older adults at risk. Consequently, it is recommended that the TUG be applied in combination with other functional tests and validated questionnaires, thereby increasing diagnostic accuracy and clinical utility in fall prevention [36].

The Timed Up and Go Test (TUG) is a widely used tool to assess fall risk in older adults. However, in its traditional form, based solely on the total time taken to complete the test, it shows important limitations in its predictive capacity for fall risk, with an average accuracy of only 54.2%. In contrast, the integration of inertial measurement units (IMUs) during the execution of the TUG enables the extraction of additional gait and balance parameters, which significantly enhances its performance and increases predictive accuracy to 79.7% [37]. Moreover, although the TUG applied in isolation shows reduced sensitivity and specificity, when complemented with sociodemographic variables (such as sex, joint pain, or visual problems) and fall risk questionnaires, it yields more robust predictive models with greater clinical relevance. This approach is considered an ideal initial strategy in the community, as it enables the early identification of individuals at risk [38].

We believe that this study may contribute to fall risk prevention, provide insights into foot biomechanics, and improve quality of life while reducing morbidity and mortality in the elderly population. Our hypothesis was that individuals with FHL would exhibit a higher risk of falls compared to those without this condition. Thus, we proposed the following research question: Does functional hallux limitus increase the risk of falls in older adults?

## 2. Materials and Methods

### 2.1. Sample Size

Sample size calculation was conducted using G*Power 3.1.9.7 software, applying t-test family assessments and correlation statistical tests. Parameters included a normal distribution, two-tailed hypothesis, an effect size of 0.50, an α error probability of 0.05, and a power of 80% (β = 20%). Based on these values, the required sample size was 38 participants, but finally a total of 40 participants were recruited.

This study was conducted as a case–control design, in accordance with the Declaration of Helsinki and human research regulations [39]. It followed the guidelines established by the “Strengthening the Reporting of Observational Studies in Epidemiology (STROBE)” initiative [40]. The protocol was approved by the Ethics Committee of the University of León, Castilla y León, Spain (internal registration number: ÉTICA-ULE-035-2025), and informed consent was obtained from all participants prior to inclusion.

Exclusion criteria included the presence of recent severe ligament injuries, history of surgery, bone fractures, muscle injuries in the lower limbs, abnormal gait patterns, medical restrictions on exercise, or any other health condition that could negatively influence study outcomes. The inclusion criteria were patients over 65 years of age, were either asymptomatic FHL patients or non-FHL subjects (control group), and who were recruited from those attending the University Podiatry Clinic at the University of León.

#### 2.1.1. FHL Patient Group

Twenty asymptomatic patients with FHL were recruited, presenting a range of motion (ROM) of the first metatarsophalangeal joint weight-bearing of less than 60°, even though 60–75° is considered necessary for normal human gait, as described by Dananberg [24,25,41]. Specifically, the selected patients had an ROM of less than 12° in static bipedal stance. All patients were clinically evaluated using an analog goniometer [42,43,44] and a podoscope. A podoscope was also employed, not as a diagnostic device for FHL, but as a support platform that elevated the patient from the ground and facilitated goniometric measurements.

#### 2.1.2. Control Group

Twenty patients without FHL were recruited, all with a ROM of the first metatarsophalangeal joint greater than 60°, both in weight-bearing and non–weight-bearing conditions. They had no history of injuries or dysfunction in the feet or lower limbs and reported no pain in those areas. Pre-study evaluations consisted of a patient anamnesis, review of medical history to rule out diseases or pathologies that could affect gait patterns, and observation of normal gait over a 10 m walkway.

### 2.2. Gait Mobility Assessment and Measurement Instrument

To assess gait mobility, the validated and instrumented TUG test was applied [32]. The test involves seating the participant on a chair (approximately 44 cm high for this study), with hands on their thighs and not leaning against the backrest. The participant then stands up without using their arms, walks 3 m along a firm-surfaced hallway, turns 180°, and sits down again. This test is widely recognized as an effective method for measuring both walking speed and dynamic balance, with a cutoff time of ≥13.5 s used to identify individuals at greater risk of falling [45,46].

The measurement instrument used to assess gait parameters and events during the TUG test was the Wiva Science sensor (Wiva Science-LetSense Srl, Bologna, Italy) (Figure 1). The validated WIVA SCIENCE IMU is based on the technology of its predecessor prototype, the F4A. Accelerometer and gyroscope data were recorded at a frequency of 100 Hz, with sensitivities of ±1.5 G and ±300°/s, respectively. The device was positioned at the sacral region, aligned with the anatomical axes of the pelvis. Prior to dynamic acquisition, a five-second static calibration in bipedal stance was performed to obtain reference angles. Finally, the signals were filtered using a fourth-order Butterworth filter with a cut-off frequency of 8 Hz [47]. (Figure 2). Gait parameters from the IMU sensors [48] were extracted using Biomech software (version 1.6.1.14687, LetSense Group Srl, Bologna, Italy).

The variables studied were as follows: Step rate (steps/min), Gait cycle duration (s), Right step duration (s), Left step duration (s), Swing phase duration (%), Right foot swing phase duration (%), Left foot swing phase duration (%), Stance phase duration (%), Right foot stance phase duration (%), Left foot stance phase duration (%), Right step length (cm), and Left step length (cm).

### 2.3. Postural Stability Analysis

The Berg Balance Scale (BBS) is a validated tool [30,31] used to assess balance through 14 functional tasks performed in a pre-established order: sitting to standing, standing unsupported, sitting unsupported, standing to sitting, transfers, standing with eyes closed, standing with feet together, reaching forward with an outstretched arm, picking up an object from the floor, looking behind over shoulders, turning 360°, placing alternate foot on a stool, standing with one foot in front, and standing on one leg.

Each item is scored from 0 to 4 points. Balance scores were grouped into three categories: 0–20: poor score/high fall risk; 21–40: moderate score/moderate fall risk; 41–56: good score/low fall risk [31].

### 2.4. Fear of Falling Assessment

Fear of falling was assessed using the Falls Efficacy Scale—International (FES-I), a validated questionnaire [26,27,28,29] composed of 16 items assessing concern over falling during various physical and social activities inside and outside the home. Each item is scored from 1 to 4, for a total score range of 16 to 64 points.

The items include the following: Cleaning the house, Dressing/undressing, Preparing simple meals, Taking a bath or shower, Shopping, Sitting down or getting up from a chair, Going up or down stairs, Walking outside the house, Reaching up or bending down, Answering the phone, Walking on a slippery surface, Visiting a friend or relative, Going to a crowded place, Walking on uneven surfaces, Walking uphill or downhill, and Attending a social event. A score above 20 on the FES-I suggests an increased fall risk, particularly in women aged 50 to 65 [27].

### 2.5. Statistical Analysis

Data analysis was performed using SPSS version 22 for Windows (SPSS Inc., Chicago, IL, USA). The Shapiro–Wilk test was applied to assess the normality of the data distribution. Since the data did not meet the assumptions of normality, the Mann–Whitney U test was selected to compare differences between the FHL and control (C) groups. A significance level of *p* < 0.05 was considered for all analyses.

## 3. Results

A total of 40 participants were included in the study. There were eight men and thirty-two women, with a mean age of 78.92 ± 8.98 years, mean body mass of 66.82 ± 12.11 kg, mean height of 160.02 ± 7.37 cm, and mean body mass index (BMI) of 26.06 ± 4.04. As shown in Table 1, No statistically significant differences were found in body mass, height, or BMI (kg/m^2^) [49].

### 3.1. Analysis of Gait Mobility, Postural Stability, and Fear of Falling

As shown in Table 2, no statistically significant differences were observed between groups for the TUG scores (*p* = 0.694). Additionally, no statistically significant differences were observed between groups for the BBS (*p* = 0.903) or the FES-I scores (*p* = 0.913) (Table 2).

### 3.2. Gait Parameters

As shown in Table 3, no statistically significant differences were found between the FHL and control groups in any of the spatiotemporal gait parameters analyzed (all *p* > 0.05). Although cadence was slightly reduced and right step length slightly greater in the FHL group, while left step length was greater in the control group, these differences did not reach statistical significance.

## 4. Discussion

In the present study, we sought to address a gap in the literature regarding whether functional hallux limitus (FHL), a condition known to alter foot biomechanics, contributes to impaired balance and an increased risk of falls in older adults. Although falls represent a major public health concern in this population, there is limited evidence on the potential role of FHL in this context. Therefore, this study aimed to compare postural stability in patients with FHL to that of healthy subjects using an inertial measurement unit (IMU) and validated clinical tests.

In the present study, we compared the postural stability of patients with Functional Hallux Limitus (FHL) to that of healthy subjects using an inertial measurement unit (IMU). Our findings indicate that patients with FHL did not show significant differences in the spatiotemporal parameters of gait compared to healthy individuals, suggesting that the presence of FHL does not evidently affect postural stability under the conditions evaluated.

The scores obtained in the TUG and BBS tests did not show significant differences between the FHL group and the control group, indicating that mobility during gait and balance do not appear to be compromised in patients with FHL. These results do not support our initial hypothesis that individuals with FHL are at a higher risk of falls compared to those without FHL.

Previous studies have demonstrated a direct relationship between foot and ankle factors and fall risk, including ankle flexibility, plantar sensitivity, toe flexor strength, foot pain [50], deformities of the first toe such as hallux valgus, and deformities of the lesser toes such as hammer toes and claw toes [51]. However, based on the results of this study, we cannot conclude that asymptomatic FHL is a risk factor for falls.

It is worth highlighting the research by Dananberg [24], which suggests that FHL delays heel lift, potentially altering the propulsive phase of gait, which could impact long-term stability. Furthermore, FHL interferes with the activation of the Windlass Effect, reducing the structural efficiency of the foot and leading to compensatory patterns that may not be detectable through standard evaluations.

In addition, the results of our study differ from those published by Dananberg [25], which demonstrate that the absence of heel lift during the single support phase represents a progression of delayed heel rise, significantly affecting gait mechanics. This phenomenon is observed more frequently in older individuals, where reduced mobility and stability contribute to less efficient gait, causing an increase in double support time, which, in turn, compromises dynamic balance and increases the risk of instability.

Moreover, discrepancies were observed between our results and those found in other studies, in which patients with hallux valgus (HV) showed significant alterations in the same parameters [18]. In that study, the HV group recorded a mean TUG score of 9.1 ± 1.4 s, significantly greater than the control group (7.8 ± 2.1 s, *p* = 0.002), indicating reduced functional mobility and greater difficulty completing the test. Additionally, patients with HV presented a mean BBS score of 51.9 ± 3.5 points, significantly lower than that of the control group (55.7 ± 0.4 points, *p* < 0.001), reflecting greater impairment in postural stability and balance control in this group.

Finally, the FES-I was notably greater in patients with HV (24.4 ± 5.8 points versus 18.0 ± 2.4 points in the control group, *p* < 0.001), indicating an increased fear of falling in this group of patients. In contrast, our results showed no significant differences in FES-I scores between the control and FHL groups.

The difference in outcomes between both studies suggests that, while HV negatively affects mobility, balance, and fear of falling, FHL does not appear to significantly impact these parameters under the evaluated conditions. In the case of HV, the structural alteration and deviation of the first metatarsal may compromise postural stability, which is reflected in increased TUG scores, lower BBS scores, and greater fear of falling according to the FES-I. Conversely, in asymptomatic FHL, the restriction in dorsiflexion of the first metatarsophalangeal joint does not seem to significantly alter postural control or confidence in gait.

Likewise, patients with FHL in our study did not present significant differences in mobility, balance, or perceived fear of falling in a controlled environment and along a firm-surfaced hallway. However, previous research suggests that these differences may become evident in more demanding environments or situations requiring rapid postural adaptations, such as walking on uneven surfaces, supporting the hypothesis that certain biomechanical deficits may not be evident in standard tests like the TUG or BBS, but may emerge under conditions of greater postural demand [52].

Another factor to consider is that the participants in our study presented with FHL without pain. However, other studies have shown that disabling foot pain is associated with a greater incidence of falls [53], likely due to changes in gait and reduced balance. This suggests that, although limited hallux mobility may influence foot mechanics, the determining factor in fall risk may be the presence of pain during mobilization of the first metatarsophalangeal joint.

Furthermore, subsequent studies have shown that the traditional TUG test only measures the total time required to complete the test, which limits its ability to detect specific alterations in mobility and balance [32]. However, when using a sensor alongside the TUG, the test can be analyzed in different phases, offering greater sensitivity for postural control and identifying subtle gait alterations, making it a superior tool for evaluating balance and fall risk [54,55,56]. For these reasons, the TUG test accompanied by an IMU was selected in our study to provide a more detailed analysis of gait stability and functionality in patients with FHL.

As limitations of this study, it should be noted that many elderly participants became fatigued after completing the BBS and TUG tests, which may have affected the scores obtained in participants with greater physical and cognitive deterioration. Additionally, we must mention the difficulty of the WIVA IMU in accurately capturing data at reduced walking speeds, requiring repetition of the TUG test in some subjects. However, it is important to note that this is a common issue found with this type of sensor [57].

## 5. Conclusions

The results of this study suggest that asymptomatic functional hallux limitus (FHL) does not significantly affect mobility, postural stability, or fear of falling in older adults.

No statistically significant differences were found in the results of the Timed Up and Go Test (TUG), the Berg Balance Scale (BBS), or the Falls Efficacy Scale—International (FES-I) between the FHL and control groups. Similarly, no significant differences were observed in the spatiotemporal gait parameters analyzed using an inertial measurement unit.

These findings indicate that asymptomatic FHL does not appear to be a determining factor in fall risk under controlled clinical conditions. Nevertheless, further research is needed in ecological environments, as well as in populations with painful FHL or additional comorbidities, to better understand the functional implications of this condition.

## Figures and Tables

**Figure 1 bioengineering-12-01094-f001:**
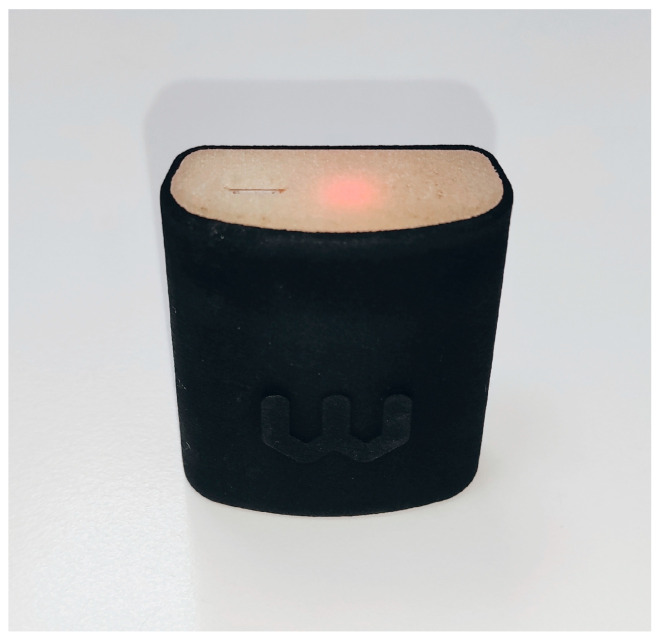
Inertial Measurement Unit Wiva Science.

**Figure 2 bioengineering-12-01094-f002:**
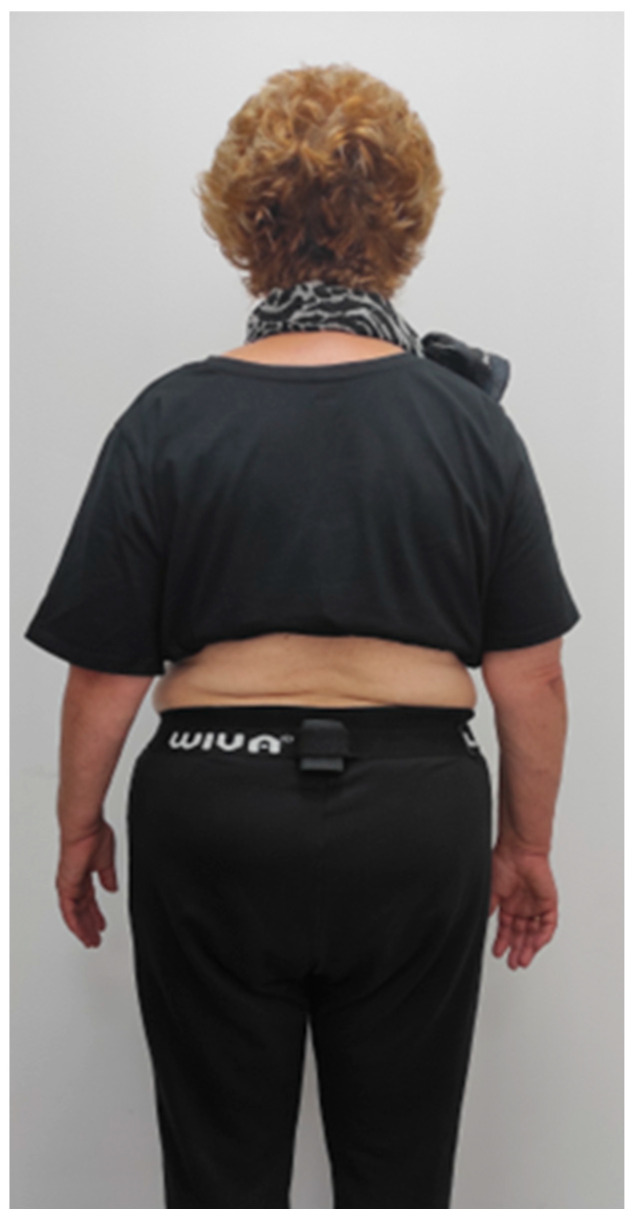
Inertial Measurement Unit Wiva Science location.

**Table 1 bioengineering-12-01094-t001:** Demographic characteristics of the control and functional Hallux limitus groups.

Variable	Control Group Mean ± SD (IC95%)	FHL Group Mean ± SD (IC95%)	*p* Value
Age (years)	81.20 ± 7.92 (73.27–84.67)	76.65 ± 9.59 (72.44–80.85)	0.110
Height (cm)	159.7 ± 7.99 (151.70–163.20)	160.35 ± 6.89 (157.32–163.37)	0.784
Body mass (kg)	68.30 ± 11.11 (57.18–73.17)	65.35 ± 13.16 (59.57–71.12)	0.448
BMI (kg/m^2^)	26.71 ± 3.32 (23.38–28.16)	25.41 ± 4.64 (23.37–27.45)	0.317

The values are shown as the mean and standard deviation, with the range in parentheses. The Mann–Whitney U test was used for statistical analysis, with the significance level set at *p* < 0.05.

**Table 2 bioengineering-12-01094-t002:** Evaluation of walking mobility, balance and fear of falling.

Variable	Control Group Mean ± SD (IC95%)	FHL Group Mean ± SD (IC95%)	*p* Value
TUG (s)	14.09 ± 4.25 (9.83–15.95)	14.04 ± 6.140 (11.348–16.731)	0.694
VALUE BERG (points)	35.70 ± 14.30 (21.39–41.96)	35 ± 14.991 (28.429–41.570)	0.903
FES (points)	27.25 ± 14.26 (12.98–33.50)	27.35 ± 11.944 (22.115–32.584)	0.913

TUG: Timed Up and Go test; BBS: Berg Balance Scale; FES: Falls Efficacy Scale; C group: control group; FHL group: functional hallux limitus group. The values are shown as the mean and standard deviation, with the range in parentheses. The Mann–Whitney U test was used for statistical analysis, with the significance level set at *p* < 0.05.

**Table 3 bioengineering-12-01094-t003:** Spatiotemporal gait parameters (mean, standard deviation) for control and FHL groups.

Variable	Control Group		FHL Group		*p* Value
	Mean ± SD (IC95%)	Median (IR)	Mean ± SD (IC95%)	Median (IR)	
Step rate (step/min)	50.15 ± 6.68 (47.02–53.27)	51.4 (47.92–53.89)	48.91 ± 6.07 (46.06–51.75)	49.65 (47.23–50.73)	0.473
Gait Cycle Duration (s)	1.21 ± 0.21 (1.11–1.31)	1.15 (1.11–1.25)	1.24 ± 0.21	1.20 (1.11–1.27)	0.551
Right Step Duration (s)	0.61 ± 0.10 (0.55–0.66)	0.58 (0.56–0.63)	0.63 ± 0.10	0.60 (0.58–0.66)	0.266
Left Step Duration (s)	0.61 ± 0.10 (0.56–0.66)	0.59 (0.56–0.63)	0.61 ± 0.12 (0.55–0.67)	0.60 (0.55–0.62)	0.684
Swing Phase Duration (%)	25.01 ± 2.41 (23.88–26.14)	24.95 (24.10–25.79)	24.99 ± 1.79 (24.16–25.83)	25.15 (24.16–26.68)	0.532
Right Foot Swing Phase Duration (%)	25.19 ± 3.34 (23.62–26.75)	25.15 (23.81–25.78)	24.11 ± 2.92 (22.73–25.48)	24.30 (23.23–26.36)	0.371
Left Foot Swing Phase Duration (%)	24.93 ± 2.42 (23.79–26.07)	25.05 (24.21–26.04)	25.06 ± 2.70 (23.79–26.33)	25.45 (24.13–26.79)	0.704
Stance Phase Duration (%)	74.98 ± 2.41 (73.85–76.11)	75.05 (74.20–75.90)	75.00 ± 1.79 (74.16–75.83)	74.85 (73.31–75.83)	0.532
Right Foot Stance Phase Duration (%)	74.81 ± 3.34 (73.24–76.37)	74.85 (74.21–76.18)	75.89 ± 2.92 (74.51–77.26)	75.70 (73.63–76.76)	0.371
Left Foot Stance Phase Duration (%)	75.06 ± 2.42 (73.92–76.20)	74.95 (73.95–75.78)	74.93 ± 2.70 (73.66–76.20)	74.55 (73.20–75.86)	0.704
Right Step Length (cm)	48.28 ± 3.80 (46.49–50.06)	47.85 (46.10–50.89)	50.77 ± 6.36 (47.79–53.75)	52.60 (46.78–55.08)	0.119
Left Step Length (cm)	51.57 ± 3.72 (49.82–53.31)	52.15 (49.10–53.71)	49.22 ± 6.36 (46.24–52.20)	47.40 (44.91–53.21)	0.14

FHL group: Functional Hallux Limitus group. The values are shown as the mean and standard deviation with the 95% confidence interval and median and interquartile range (IR) in parentheses. The Mann–Whitney U test was used for statistical analysis.

## Data Availability

The data from this study will be available upon reasonable request to the corresponding author.

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
