# Peer review of "Risk of Fall in Patients with Functional Hallux Limitus: A Case–Control Study Using an Inertial Measurement Unit"

_bioengineering, 2025, doi:10.3390/bioengineering12101094_

Round 1

Reviewer 1 Report

Comments and Suggestions for Authors

This study will contribute to fall risk prevention, provide insights into foot biomechanics, and improve quality of life while reducing morbidity and mortality in the elderly population. It is attractively interesting, so accepting in this version is suggested.

Author Response

"We sincerely appreciate the reviewer’s evaluation and his positive comments. We are very grateful for recognizing the potential impact of our study on fall risk prevention and quality of life in the elderly population, as this has encouraged and motivated us to continue our research. We are pleased to know that the manuscript is considered suitable for acceptance in its current version."

Reviewer 2 Report

Comments and Suggestions for Authors

1) Real-world falls typically occur on uneven terrain, in low light, or during dual-task conditions; however, the experiment was conducted on a firm, well-lit hallway (Line 115), a setting that diverges from testing demands.
2) The inertial unit was secured solely at the sacrum (Fig. 2). Consequently, sacral accelerometry is used to infer whole-body fall risk while fore-foot kinematics remain unmeasured. This undermines mechanistic interpretation.
3) In section 2.1, "Healthy" controls are merely defined by ROM, many elderly exhibit restricted hallux motion despite passive ROM >60°. A dynamical assessment is required to label a joint truly  unrestricted.
4) Mean age is 78.92 y (line179). Even within the 65+ demographic, fall risk rises sharply. Age must be anlyzed as a covariate to determine whether the risk stems from hallux mechanics or from age imbalance within the cohort.
5) Given that 80 % of participants are female (32 F / 8 M), the manuscript must include gender-stratified analyses to determine whether FHL exhibits differential fall risk across genders.

Author Response

First of all, we would like to sincerely thank you for your comments and valuable feedback, which are of great importance to us not only for this work but also for future research.

1. Experimental setting
We chose to conduct the study in a controlled environment in order to minimize potential measurement errors from the inertial unit, since our primary aim was to accurately capture the spatiotemporal gait parameters of the participants.

2. Sensor placement
The sensor was positioned according to the manufacturer’s recommendations, as it is configured to operate in that location when the TUG mode is activated. Nevertheless, we are currently developing new lines of research with other types of sensors and alternative anatomical placements.

3. Definition of healthy controls
Participants were assessed using a podoscope and a goniometer to evaluate the range of motion. However, we acknowledge that we did not assess the first metatarsophalangeal joint dynamically. We find your suggestion very valuable and will take it into account in future studies.

4. Age as a covariate
We intentionally focused on an elderly population, as they are at the highest risk of falls. In parallel, we are conducting further research to determine the specific age ranges at which fall risk increases most sharply, including assessments in non-controlled environments.

5. Gender distribution
It was not possible to balance the number of male and female participants in the sample, which is why sex-based comparisons were not included in the demographic table.

Finally, we would like to once again express our gratitude for your thoughtful feedback. Your comments have been extremely helpful and have provided us with valuable ideas for future research directions, such as evaluating fall risk on uneven terrain using different inertial measurement units.

Reviewer 3 Report

Comments and Suggestions for Authors

The introduction needs to be more fully developed; it ends with appropriate purpose and hypothesis statements. The methods section is well organized but needs more information. The results section needs to be more concise. The discussion and conclusion was consistent with the primary purpose of the study. There are several word choice, word tense, and grammatical errors. The references need to be check to make sure they consistently follow journal instructions. See specific comments in the pdf.

Comments on the Quality of English Language

An expert English grammar is recommended to improve manuscript correctness and coherence.

Author Response

Dear Reviewer,

First of all, we would like to sincerely thank you for the time and effort you have devoted to reviewing our manuscript and for providing us with your valuable suggestions and corrections. Thanks to your observations, we have been able to address errors and improve the overall quality of the submitted work. To facilitate reading, we have marked the modified text in red.

With regard to the changes made:

  1. We have expanded the introduction, as it was initially too brief.
  2. The results section has been revised and restructured to ensure a more concise and appropriate presentation.
  3. Grammar, verb tenses, and word usage have been reviewed following your recommendations, with the additional support of external language reviewers.
  4. The references have been thoroughly revised in accordance with the journal’s guidelines, eliminating duplicates and ensuring uniform formatting.
  5. The figures have been replaced and improved to provide greater clarity and visual quality.
  6. The tables have been corrected following your suggestions, presenting the information in a clearer and more precise way.

All the authors would like to once again express our gratitude, as your contributions have directly helped to enhance the quality of this research.

Round 2

Reviewer 2 Report

Comments and Suggestions for Authors

All concerns have been addressed

Author Response

Response to Reviewer 2

We sincerely thank the reviewer for the valuable comments regarding the figures and tables. Following the recommendations, we have improved the quality of the images by converting them to 300 DPI and ensuring that each figure has a resolution greater than 1000 pixels on the longest side. In addition, we have carefully revised all figure captions to ensure their accuracy and consistency.

Furthermore, we have reviewed all the tables and standardized both the font type and size to match the rest of the text.

The modifications made are underlined in yellow.

We greatly appreciate these suggestions, which have contributed to improving the overall quality and clarity of the manuscript.

Reviewer 3 Report

Comments and Suggestions for Authors

Thank you for addressing my original comments/questions/suggestions. Several minor word choice and grammatical errors, e.g., one-sentence paragraphs (several were highlighted), remain. See specific comments in the pdf.

Comments on the Quality of English Language

Several areas where one-sentence paragraphs are used need to be corrected. This requires the authors to improve the coherence and flow of the manuscript.

Author Response

RESPONSE TO REVIEWER 3
1. I am sure I commented on one-sentence paragraphs before....these are basic
grammatical errors that need to be corrected.
RESPONSE:
We sincerely thank the reviewer for this valuable observation. We have carefully
revised the manuscript to avoid one-sentence paragraphs and adjusted the text to
improve readability and clarity throughout the document.
2. This suggests more than one publication yet you cite only one, i.e., #43; please
include other citations to support your claim or reword this sentence.
RESPONSE:
We have revised the text to ensure coherence with citation 43 and adjusted the
wording accordingly.
3. Suggest the word "may"; here is another one-sentence paragraph
RESPONSE:
We thank the reviewer for this suggestion and have proceeded to make the
corresponding change in the manuscript.
4. Suggest the word "reduced" or "less"
RESPONSE:
We thank the reviewer for this suggestion and have proceeded to make the
corresponding change in the manuscript.
